# A galactic microquasar mimicking winged radio galaxies

Josep Martí [1], Pedro L. Luque-Escamilla[2], Valentí Bosch-Ramon[3] & Josep M. Paredes[3]

A subclass of extragalactic radio sources known as winged radio galaxies has puzzled astronomers for many years. The wing features are detected at radio wavelengths as low-surface-brightness radio lobes that are clearly misaligned with respect to the main lobe axis. Different models compete to account for these peculiar structures. Here, we report observational evidence that the parsec-scale radio jets in the Galactic microquasar GRS 1758-258 give rise to a Z-shaped radio emission strongly reminiscent of the X and Z-shaped morphologies found in winged radio galaxies. This is the first time that such extended emission features are observed in a microquasar, providing a new analogy for its extragalactic relatives. From our observations, we can clearly favour the hydrodynamic backflow interpretation against other possible wing formation scenarios. Assuming that physical processes are similar, we can extrapolate this conclusion and suggest that this mechanism could also be at work in many extragalactic cases.

[1] Departamento de Física, Escuela Politécnica Superior de Jaén, Universidad de Jaén, Campus Las Lagunillas s/n, A3, 23071 Jaén, Spain. [2] Departamento de Ingeniería Mecánica y Minera, Escuela Politécnica Superior de Jaén, Universidad de Jaén, Campus Las Lagunillas s/n, A3, 23071 Jaén, Spain. [3] Departament de Física Quàntica i Astrofísica, Institut de Ciències del Cosmos (ICCUB), Universitat de Barcelona, IEEC-UB, Martí i Franquès 1, E-08028 Barcelona, Spain. Correspondence and requests for materials should be addressed to J.M. (email: jmarti@ujaen.es)

A not-so-large but significant number of low-luminosity radio galaxies exhibit extended radio lobes with an additional pair of large, low-brightness wings oriented at some angle from the jet. This gives these extragalactic sources the X or Z-shape that led to their nickname of winged radio galaxies (WRGs). Historical analysis of a scarce sample of 3C sources[1] has already revealed that more than 60% of the more powerful, edge-brightened, radio galaxies of FRII-type[2] exhibit some kind of distortion in their large-scale structure, of which 23% were found to have anti-symmetric wings. The population of WRGs has continued to increase by nearly an order of magnitude in most recent works[3, 4], with no definitive example of FRI-type galaxies being reported[5]. These sources have recently received much attention because they could be signposts of the coalescence of supermassive black holes in galaxy mergers. According to spin-flip models[6–8], the merger process leads to a change in the rotation axis of a supermassive black hole producing the jet to which the WRG morphology is attributed. Moreover, their abundance might be useful for predicting the magnitude of the gravitational wave background[4].

However, spin-flip is not the only possible explanation for origin of the wings. These features may still be relic emissions from previous orientations of a jet suffering conical precession[7] or an apparently double bipolar jet system associated with a pair of unresolved active galactic nuclei[9, 10] (AGNs). On the other hand, some authors have claimed that the wings originate hydrodynamically from the diversion of the backflow of shocked jet material along the steepest pressure gradient of the surrounding medium, rising buoyantly or being driven in this direction[1, 11–15]. It has been stated that subsonically moving wings of hydrodynamic origin should appear shorter than supersonically advancing primary lobes, although this is contradicted by observations[7]. To circumvent this issue, wings could be accelerated out of an overpressured cocoon, via a de Laval nozzle, to form a pair of loosely collimated supersonic flows of synchrotron plasma[12, 16]. The Z-shape of some WRGs, which cannot be naturally explained from spin-flip models, has been modelled as the interaction of the jets with a rotational medium induced by galaxy mergers[17, 18]. This model was later generalised to a jet-shell scenario[15]. All these theories have drawbacks and merits, and no consensus has yet been reached regarding the onset of the WRG morphology.

A key aspect here is whether the WRG shape somehow originates from the final interaction between the jets and their environment. In this case, one expects that the underlying physics will be hydrodynamical, being mainly dependent on a few basic parameters, such as the jet density, medium density and jet power, among others. The relationship between these parameters is anticipated to follow a dynamic similarity at different scales. This jet/medium interaction hypothesis would become strongly supported if WRG-analogues were found to exist in downscaled systems with bipolar relativistic jets, such as stellar microquasars[19, 20]. These systems are well known for the strong parallelism of their accretion-ejection phenomena with radio galaxies and AGNs[21, 22]. By appropriately scaling the dynamic similarity laws of fluid mechanics to the very different environments of microquasar and extragalactic jet sources, there is no a priori argument against this parallelism also applying to regions far beyond the central engine of these systems where the lobes form.

In this context, here, we report that the Galactic microquasar GRS 1758-258 has a large-scale Z-shaped morphology that mimics that of many extragalactic WRGs. This object, which has this previously unseen behaviour, is one of the two strongest hard X-ray sources in the vicinity of the Galactic Centre[23]. Its arcmin radio jets change in morphology over a matter of a few years, and based on causality arguments the source cannot be located

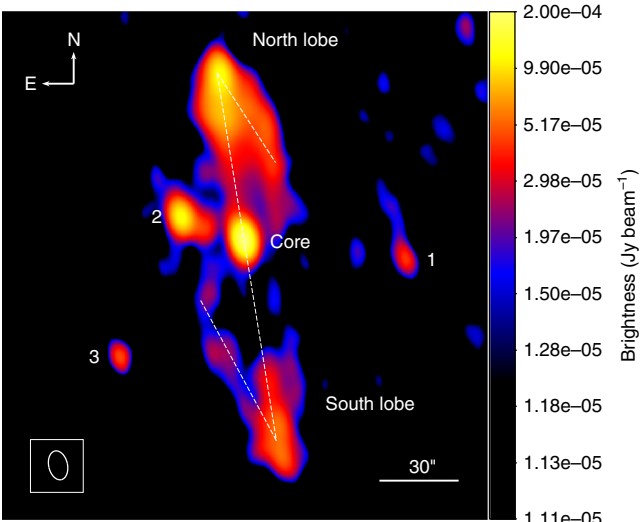

**Fig. 1** Z-shaped radio morphology of the microquasar GRS 1758-258. This map was obtained from the concatenation of VLA runs carried out at the 6 cm wavelength in the D and C array configurations conducted in 1992, 1993, 1997 (archival data) and 2016 (new data from this work). Details are given in the Methods section and Table 1. The central core and the north and south radio lobes are labelled together with the unrelated sources #1, #2 and #3 in the same field of view. The white dashed line outlines the Z-shaped appearance of the faint emission extensions emanating from both lobes. The bottom left ellipse shows the angular resolution and corresponds to the full width at half maximum of the VLA synthesised beam (11.31 × 7.18 arcsec², with a position angle of 13°). The right horizontal bar (30 arcsec long) provides the map angular scale, with north being up and east being left. The brightness levels are illustrated by the colour bar to the right. The intensity scale used is logarithmic and expressed in Jansky units (1 Jy = $10^{-26}$ W m$^{-2}$ Hz$^{-1}$). Meaningful radio emission starts at the 4σ level of approximately 15 μJy

beyond 12 kpc[24]. A heavily absorbed optical and infra-red counterpart is consistent with an intermediate-mass binary likely hosting an A-type main sequence star at a distance similar to that of the Galactic Centre, and a compact stellar companion, i.e. either an accreting neutron star or a black hole[25]. A distance value of 8.5 kpc will be adopted throughout this work. This ensures consistency with both the previous upper limit and the spectral energy distribution fits based on optical and near-infra-red observations made using the Gran Telescopio CANARIAS and the Hubble Space Telescope, respectively[25, 26]. As evidenced in the following sections, the Z-shaped structures in GRS 1758-258 allow us to strengthen the plausibility of the wing backflow scenario over alternative interpretations proposed for its extragalactic analogues at a larger scale.

## Results

**Interferometric radio observations.** Using the Karl G. Jansky Very Large Array (VLA), in 2016, we conducted a new observation of GRS 1758-258, which had remained unmonitored by sensitive radio interferometers for nearly a decade. The map in Fig. 1, resulting from the combined analysis of our new campaign and selected sensitive VLA archival radio observations (see Methods section and Table 1), provides the current deepest view of this microquasar environment at cm radio wavelengths.

**Discovery of a Z-symmetric radio morphology in GRS 1758-258.** Extended emission features, emanating from the northern and southern radio lobes, can clearly be seen in Fig. 1. These features were marginally detected in previous works[24, 27], being interpreted as a surrounding cocoon-like structure made of

**Table 1 Log of VLA 6 cm observations used in this work**

| Project code | VLA conf. | Observation date | On-source time (s) | Bandwidth (MHz) | rms Noise (μJy/beam) | Relative weight[a] |
|---|---|---|---|---|---|---|
| 16A-005 | C | 2016 Mar 04–22 | 7659 | 2048 | 4.3 | 75% |
| AS930 | C | 2008 Apr 01–12 | 14,505 | 50 | 9.4 | [b] |
| AM560 | C | 1997 Aug 03–24 | 20,810 | 50 | 9.2 | 17% |
| AM428 | CD | 1993 Oct 03–04 | 6460 | 50 | 11.8 | 8% |
| AM345 | D | 1992 Sep 26–27 | 5690 | 50 | | |
| AM385 | D | 1992 Sep 10–11 | 8550 | 50 | | |

[a] Weight indicative of contribution to final image shown in Fig. 1. For CD-D configuration data, the stated value is for the three available projects combined
[b] Not used for imaging in Fig. 1

shocked jet matter originating from the interaction of the jet with its interstellar medium (ISM). With the new, more sensitive data, the faintest southern jet wing is detected well above the 6σ level (see also the contour map in Supplementary Fig. 1). Now, the extended emission complex actually exhibits the Z-symmetric appearance typical of some WRGs. The observed morphology also departs significantly from the one-sided, elliptical ring-like feature known to surround the microquasar Cygnus X-1 and probably inflated by a dark jet[28]. A few radio sources that overlap with the field of view (numbered #1, #2 and #3 in Fig. 1) are also present. They remain detected even in super-resolution maps with pure uniform weight. This observation clearly indicates a compact nature; therefore, these radio sources are likely to be unrelated to the extended jet flow.

## Discussion

We have carried out a first assessment of the different WRG scenarios mentioned above for the case of GRS 1758-258, although a full theoretical approach is beyond the scope of this work. A Z-shape is hard to reconcile with a spin-flip model as there is no emission connecting the wings to the central source[29]. In addition, according to our current understanding of stellar evolution, black hole mergers are certainly unexpected to occur during the GRS 1758-258 jet lifetime. Moreover, assuming that both the accretion rate and disk viscosity properties remain constant, the jet realignment time scale[30, 31] would be on the order of

$$\frac{\tau_{\text{alig}}}{\text{Myr}} \sim 0.3 \left(\frac{a}{0.1}\right)^{11/16} \left(\frac{\alpha_{\text{disk}}}{0.03}\right)^{13/8} \left(\frac{M_{\text{BH}}}{10^8 M_\odot}\right)^{-1/16} \left(\frac{\dot{M}_{\text{acc}}}{0.1 \dot{M}_{\text{Edd}}}\right)^{-7/8},$$

(1)

with $a$ being the non-dimensional spin of the black hole, $\alpha_{\text{disk}}$ the disk viscosity parameter, $M_{\text{BH}}$ the black hole mass and $\dot{M}_{\text{acc}}$ the rate of accretion. For a typical AGN (i.e. 3C 293), $\tau_{\text{alig}} \sim 0.7$ Myr, but for a microquasar black hole with $10M_\odot$, $\tau_{\text{alig}} \sim 1.3$ Myr, which is completely unrealistic, as it considerably exceeds the jet travel time and age estimated below (within $10^3$–$10^5$ year).

Neither precessing models are appropriate to explain the Z-type morphology in the GRS 1758-258 case. At first glance, jet precession-like behaviour (i.e. actual precession or pseudoprecession caused by Kelvin–Helmholtz instabilities) could occur in this system given the noticeable position changes in the most conspicuous northern hot spot (Fig. 2). These motions develop on a few year time scale and are compatible with those previously reported[24]. From the central core point of view, the observed hot spot shifts imply a precession-like cone angle of, at most, a few degrees. This consideration seems difficult to reconcile with the fact that the GRS 1758-258 wings are extremely long features. Their projected arc-min length is almost equal to that of the primary lobes themselves, equivalent to approximately 3 pc at an assumed Galactic Centre distance of 8.5 kpc. Invoking precession

to explain these features would require a much larger cone angle of several tens of degrees. However, to our knowledge no such significant changes in the position angles of these jets have ever been reported.

Another possibility would be a change in the accretion disc long-term configuration, leading to a change in the jet direction. Independent of the jet formation mechanism (i.e. the Blandford–Payne[32] vs. the Blandford–Znajek[33] mechanism), the large-scale jet orientation depends on the global accretion disc orientation[34], which in turn depends on the accreted material angular momentum. This change in the accretion disc orientation requires a strong modification in the star-compact object mass transfer on time scales between those of the backflow and of the overall large-scale dynamics ($\sim 10^3$–$10^5$ year; see below). Such a marked change, if possible at all, would point to an evolved stellar companion, which is more prone to significant mass-loss changes than an ordinary main sequence star.

All the previous reasoning arguably leaves a hydrodynamical interpretation as the only and most logical scenario. The observed facts also point in this direction. The projected length of the southern jet is 30% longer than that of the northern one; the brighter northern jet and hot spot are closer to the central core than their fainter southern counterparts. This is consistent with these jets evolving in an inhomogeneous ISM. Continuous jets propagating in a medium follow an evolution in terms of length where $l_{\text{jet}}(t) \sim (P_{\text{jet}}/\rho)^{1/5} t^{3/5}$ with $t$ being the time, $P_{\text{jet}}$ the jet power, and $\rho$ the medium density[35, 36]. With all the remaining conditions being equal for both jets, the observed 30% difference in $l_{\text{jet}}$ would correspond to a north-south density ratio of $(l_{\text{jetS}}/l_{\text{jetN}})^5 \sim (1.3)^5 \sim 4$. On the other hand, although the brighter northern jet emission could be associated with Doppler boosting effects, as the jet would be approaching the observer, this cannot be the case for the brighter emission of the northern backflow (assuming that it is fast enough to relativistically beam its emission), as its motion should be directed away from the observer. A brighter northern backflow is naturally explained if the proposed higher density of the ISM gives rise to a more compact interaction region, with a higher magnetic field, and thus stronger synchrotron radio emission.

Inspection of the single dish surveys of CO emission in the Galactic Plane[37] strongly supports the idea of a north-south density difference, as illustrated by Fig. 3 and Supplementary Figs. 2 and 3. A conspicuous gas cloud is found just north of GRS 1758-258, at a radial velocity corresponding to a distance of 8.5 kpc based on kinematic arguments[38], with the microquasar practically lying at its outer edge. This location, and the likely coincidence in distance, indicates a real physical interaction between the two objects. In this context, the evolution of the northern jet would be significantly slowed down as it impinges the cloud, whereas the southern jet would flow towards a not-so-dense environment, finding less resistance than the northern jet. For the same reason, the backflow from the northern jet would become the longest winged feature as it flows towards the south.

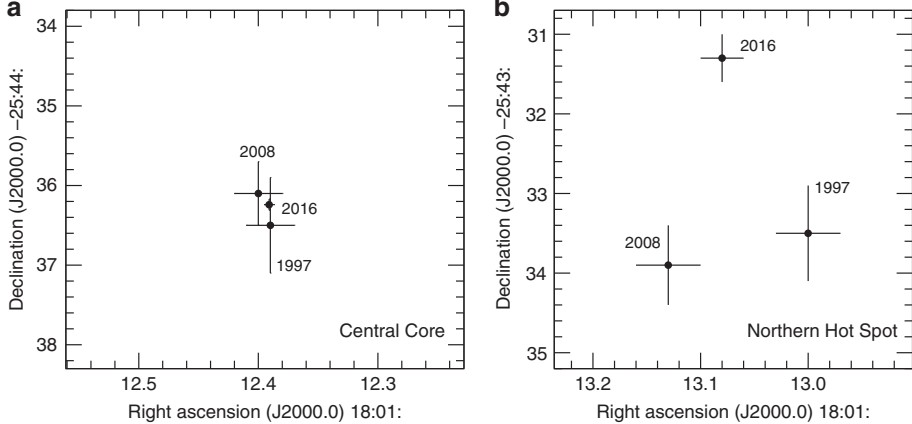

**Fig. 2** Comparison of the central core and northern hot spot positions in GRS 1758-258. The equatorial coordinates of these microquasar features during selected epochs of VLA observations are displayed in the J2000.0 reference system. The **a** and **b** panels correspond to the central core and the northern hot spot, respectively. Although the central core remains fixed within astrometric errors, the hot spot experiences clear shifts on the order of a few arcsec in time scales of years. Error bars correspond to one standard deviation, as reported by the JMFIT task of AIPS

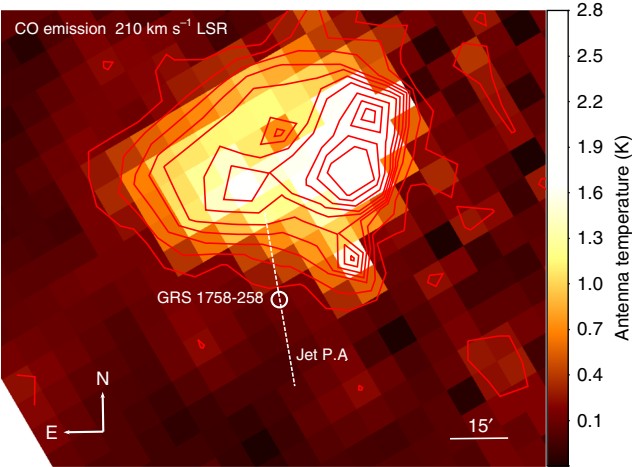

**Fig. 3** Cloud of CO along the GRS 1758-258 line of sight. This map has been created from the Dame et al. survey of CO emission in the Galactic Plane[37]. The cloud is outlined with red contours and its emission peaks at a radial velocity of 210 km s$^{-1}$ with respect to the local standard of rest (LSR), corresponding to a kinematic distance of 8.5 kpc. The position of the microquasar is marked with a white circle, and the dashed line is coincident with the position angle of the jets. The horizontal bar (15 arcmin long) gives the angular scale. North is up, and east is left. The brightness scale corresponds to the antenna temperature of CO emission in Kelvin

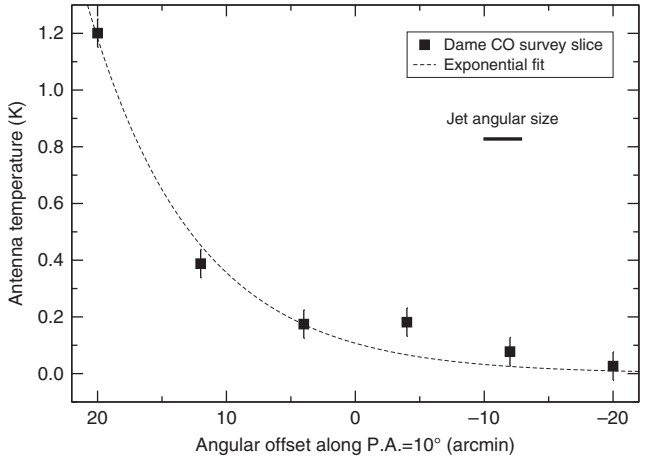

**Fig. 4** Slice of CO emission centred on GRS 1758-258. The black squares taken from the Dame survey illustrate how the CO antenna temperature changes along the position angle of the microquasar jets. Error bars are defined as one standard deviation of the emission-free background. The left side corresponds to the direction of molecular cloud impingement, whereas the right side denotes the region far away from it. The dashed line displays the exponential fit used in the main text to estimate the density variations on scales comparable to the jet angular size (3 arcmin long thick horizontal bar)

In addition, both backflow features would exhibit the north-south brightness difference mentioned above. This scenario could also naturally explain the clear backflow asymmetry with respect to the jet axis, as ISM density gradients other than the one from north to south may exist. Namely, a negative east-west density gradient at the north, and a positive one at the south, would deflect the material shocked at the jet hotspots towards the less dense region and would make the backflows point in opposite directions away from the jet axis (similar to asymmetric density distribution effects in WRGs[14, 16]). Unfortunately, the CO distribution shown in Fig. 3 does not confirm nor rule out such east-west gradient hypothesis. Its modest resolution prevents a bold statement beyond the existence of a strong north-south density gradient. Nevertheless, assuming that CO emission is optically thin at the edge of the cloud in which GRS 1758-258 is located,

the CO brightness profile would be consistent with a cloud density actually changing by the previously estimated factor of ~4 on a linear scale comparable to the extended jets. We estimated this from a simple exponential fit to the profile of CO emission along the position angle of the jets, as shown in Fig. 4. Here, the fitted antenna temperature varies by this amount within a distance of approximately three times the jet linear size (~3 arcmin). Unfortunately, the coarse angular resolution of the Dame survey does not allow a more detailed analysis beyond this simple order of magnitude assessment.

It is also worth noting that assuming $l_{jet} \sim 4$ pc, $P_{jet} \sim 10^{35}$–$10^{37}$ erg s$^{-1}$, and number density $\rho/m_H \sim 0.1$–10 cm$^{-3}$, one obtains jet ages of ~$10^4$–$10^5$ year. Longer jet ages are not favoured because they require low-power, high-density scenarios inconsistent with the undetected proper motion of the central binary system, which would significantly affect the jet shape. In fact,

Fig. 1 suggests an upper limit for jet bending of a few degrees ($\sim$0.1 rad), which implies jet ages below $\sim$4·$10^3$ year · (100 km s$^{-1}$/$v$), where $v$ is the projected proper velocity of the system. From the most recent core position in Supplementary Table 1 and previous high-angular resolution VLA observations[39], one infers that $v \lesssim 400$ km s$^{-1}$, thus constraining the jet age to be greater than $\sim 10^3$ year.

From all these results, we can state that GRS 1758-258 presents a Z-type morphology strongly resembling that of many WRGs, which can be simply explained by a light jet propagating in a medium with an asymmetric density distribution. In this context, long wings are allowed to exist, accounting for the same morphological characteristics observed in WRGs[5, 14]. In addition, 3D simulations of interactions of jets with medium inhomogeneities under suitable conditions are in agreement with the observed morphology in this microquasar[40, 41]. Assuming dynamic similarity in the jet/medium interaction, what we expect to observe in GRS 1758-258 and WRGs would be the manifestation of the same type of phenomenon; therefore, from microquasars we can extract inferences adaptable to the extragalactic case. The hydrodynamical scenario appears then as a natural interpretation of the WRG morphology with clear observational proof in a scaled-down Galactic context. Nevertheless, our finding does not necessarily exclude that other proposed scenarios for WRGs are at work in some cases.

The preferred hydrodynamic model that most closely matches our observations is the one based on a light jet/medium collision[15, 17]. The alternative scenario of overpressured cocoons is less likely, as microquasars usually evolve in low-density environments[42, 43], and no channel connecting wings with the central source is evident in our highly sensitive maps. Moreover, the strong axial asymmetries would still require inhomogeneous environments in this case. Buoyancy may have a role here, but not a protagonist one.

Therefore, our discovery strengthens the plausibility of the jet/medium interaction and backflow scenario in the genesis of Z-shaped morphologies. Beyond this robust result, understanding Z-type sources such as GRS 1758-258 might yield clues regarding the nature of X-shaped WRGs considering that a high percentage of these sources could be in fact Z-shaped ones[15, 29] located in the distant universe, being observed without enough sensitivity. Provided that dynamic similarity applies, our finding that the WRG morphology occurs in a Galactic source as a result of a clear jet/medium collision yields another possible implication in the extragalactic field. In particular, our GRS 1758-258 results lead us to speculate regarding a lower frequency of spin-flip events in the observed WRG population. One should then be cautious and not consider all wing-like features in radio galaxies as secure signposts of past merger events. If correct, this could also affect the expected gravitational wave background, whose study is beyond the scope of this paper.

One can also anticipate that additional WRG-like microquasars will possibly emerge in the near future when modern interferometers other than the existing Jansky VLA or eMERLIN, such as the Square Kilometre Array and MeerKAT, become available. Although a quantitative prediction is not straightforward, this expectation appears conceivable when considering that nearly one-third of confirmed microquasars currently known in the Galaxy[44] display compelling evidence of interaction with their ambient ISM. This includes not only co-location within a dense molecular cloud[45, 46] but also deceleration of the relativistic ejecta[47], direct jet collision with nearby clouds[48], and long-distance effects due to jet impact[49], in addition to the Cygnus X-1 jet-driven bubble mentioned above[28]. Nevertheless, for a definitive answer to the question of WRG morphology, we need further observations of sufficient quality to fully trace the lobe/wing structure and thus to have a useful characterisation of these sources to test the theoretical models. Only microquasars offer us hope of studying the live evolution of relativistic jets at large distance scales from their sources. Meanwhile, GRS 1758-258, with its bipolar jets evolving at human time scales, provides an excellent and nearby testbench for such a purpose.

## Methods

**Selection of VLA observing runs.** In our quest for better sensitivity in an attempt to reveal fainter structures around the microquasar GRS 1758-258, in this work we combined both historical and modern VLA observations. The log of the observing runs used is listed in Table 1, all of which correspond to C-band receivers at the 6 cm wavelength. The selection criterion used was to search for observing runs longer than one hour that were either continuous or accumulated within less than a week. Some 1992 observing runs satisfying this criterion were rejected because the central source was clearly undetected, which could affect the homogeneity of the combined data set. The sensitivity of modern observations, after the state-of-the-art VLA upgrade that began in 2011, is remarkably boosted with respect to the old system, especially thanks to the GHz bandwidth routinely available at present.

The VLA antenna positions were selected to be in the C configuration of the array, with maximum baselines of 3.4 km. For GRS 1758-258, the largest angular scale that can be mapped with this configuration matches well the arcmin dimensions of the jets. To further enhance the sensitivity to extended emission, we also retrieved the best data sets available for our target in the more compact D and CD configurations with the same selection criterion as stated above. The historical data sets were all acquired in two intermediate frequency bands, both of which were 50 MHz wide, sampling both the right and left circular polarisation products from the VLA correlator. In turn, our modern observing runs were acquired using an NRAO default spectroscopic mode that provided similar correlator products but for 16 spectral windows divided into 64 channels, with each window covering 128 MHz. The full bandwidth was then 2.048 GHz. Historical and modern data sets were calibrated using the AIPS and CASA software packages of NRAO, respectively. Special attention was devoted to flagging data that were bad or corrupt due to instrumental or radio frequency interference problems. Calibrator 3C286 or 3C48 was used to establish the flux density scale as well as the bandwidth calibration for data in spectral mode. Phase calibration was always determined from interleaved scans made using the NRAO calibrator J1751-253, which is located 2.1° away from our target. Phase self-calibration could be attempted for the 2016 observing run, with reasonable success, but only when solutions were averaged with a 2 min interval across the whole GHz bandwidth. Only the 2016 radio map obtained using CASA is included here for illustrative purposes (see Supplementary Fig. 4); as shown, it is very similar to previous deep images of GRS 1758-258[24]. We also checked for any systematic shifts between the CASA and AIPS maps. Based on point sources in the field, the averaged offsets did not exceed 0.07 arcsec and 0.17 arcsec in right ascension and declination, respectively. These numbers represent a small fraction of the synthesised beam and we concluded that this would not affect our conclusions.

**Merging of VLA data sets.** Combining the historical and modern data sets is by no means an easy task given their noticeably different instrumental setups. We proceeded by averaging channels in each 2016 spectral window and splitting the modern observing run into data sets, each with consecutive pairs of spectral windows. This rendered the modern and historical observations compatible and ready to be finally concatenated using the DBCON AIPS task. To combine the data sets from different project codes presented in Table 1, we adopted a reweighting criterion based on the sum of gridding weights reported by the IMAGR AIPS task. The DBCON parameters were adjusted such that each project code was weighted according to the inverse squared of its individual root square mean (rms) noise, as listed in Table 1. The last column of the table gives the final relative weight of each project in terms of its contribution to the final image in Fig. 1. The CD and D configuration projects were first separately stacked before merging with the combined C configuration runs. The oldest data sets were converted from the B1950.0 to the J2000.0 reference system using the UVFIX AIPS task. Minor shifts in the phase centre had to be accounted for in some cases, but only at the few arcsec level.

There is a price to pay for this merging process, i.e. some degree of bandwidth smearing, as data from significantly different frequencies within the C-band are being combined. Fortunately, this kind of chromatic aberration is only severe far from the phase centre; our target and its extended jets were always very close to it. The central observing frequencies were 4.8 and 5.5 GHz for historical and modern VLA data, respectively. The smearing effect can be parametrised in terms of the fractional bandwidth times the angular offset in units of the synthesised beam. The average fractional bandwidth of the whole data sets can be estimated as $\sim$1/4, and the hot spot angular offset is $\sim$6 synthesised beams. Thus, the smearing parameter becomes approximately 1.5 at the hot spot locations. The expected reduction in peak response and synthesised beam broadening in Fig. 1 should not exceed 35 and 50%, respectively[50]. With this in mind, full natural weight maps were created using IMAGR and deconvolved using the CLEAN algorithm applied to the combined data set. A taper of 25 k$\lambda$ was applied to the interferometric visibilities to better enhance the extended emission (Fig. 1).

After some tests, the 2008 observing run was dropped from the combined data set as faint extended emission could not be well imaged from it alone, and was used only for the hot spot astrometry in Fig. 2. This action is also justified because the 2008 coverage of the Fourier plane and the hot spot signal-to-noise ratio were not as good as in 1997 and 2016. The effects of Fourier plane sampling on Fig. 2 positions were further explored by taking the best 2016 map as a model and generating a simulated data set with the uv-coverage of 1997 and 2008. The core and hot spot coordinates were consistently recovered within 0.2 and 0.5 arcsec, respectively. Thus, this possible systematic error appears to be well below the > 2 arcsec position shifts displayed in Fig. 2.

Going back to the bandwidth smearing concerns, the comparison of the final image in Fig. 1 with the CASA map in Supplementary Fig. 4 provides confidence regarding the reality of the backflow features being detected. Despite their different sensitivities to extended emission, the same backflow emission features can be traced in both images. This is very reassuring, as the CASA map was computed in multifrequency synthesis mode, where no bandwidth smearing effects are expected.

The final result in Fig. 1 clearly reveals the Z-morphology in GRS 1758-258. The secondary lobes in this microquasar are likely to evolve at subrelativistic speeds on time scales of at least several decades, i.e. longer than those of the relativistic primary jets and their hotspots (~10 year). Therefore, we do not expect too much blurring; thus, their arcmin morphology can be obtained by combining observing runs separated by ~20 year.

**Data availability**. All VLA original raw data sets can be directly downloaded from the public NRAO Science Data Archive at https://archive.nrao.edu/archive/advquery.jsp. They can be searched for using the project codes listed in Table 1.

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

## Acknowledgements

The National Radio Astronomy Observatory is a facility of the National Science Foundation operated under cooperative agreement by Associated Universities, Inc. This work was supported by the Agencia Estatal de Investigación grants AYA2016-76012-C3-1-P and AYA2016-76012-C3-3-P from the Spanish Ministerio de Economía y Competitividad (MINECO), by the Consejería de Economía, Innovación, Ciencia y Empleo of Junta de Andalucía under research group FQM-322, by grant MDM-2014-0369 of the ICCUB (Unidad de Excelencia 'María de Maeztu'), and by the Catalan DEC grant 2014 SGR 86, as well as FEDER funds. V.B.-R. also acknowledges financial support from MINECO and the European Social Funds through a Ramón y Cajal fellowship. This research was also supported by the Marie Curie Career Integration Grant 321520.

## Author contributions

J.M. and P.L.-E. conducted the new VLA observations and carried out the full data processing that revealed the WRG similarities. V.B.-R. provided a theoretical framework for interpretation. J.M.P. prompted the CO survey search for interacting clouds. All authors contributed to the discussion of results.

## Additional information

**Competing interests:** The authors declare no competing financial interests.

