## [Peer Review File · Nature Communications]

Reviewers' comments:

Reviewer #1 (Remarks to the Author):

Based on the microquasar-quasar analogy it is proposed a new way to gain insight on the physics mechanisms that produce the morphology of winged radio galaxies. The reported observations of GRS 1758-258 clearly show a Z-shaped morphology similar to that found in radio galaxies, and because of the relative short time scales in the microquasar, the authors can conclude that the Z-shaped morphology in this source is produced by hydrodynamical backflows of jet shocked material propagating in an inhomogeneous environment. The subject and reported observations are of topical interest and I strongly recommend publication in Nature.

I suggest the authors briefly discuss the "likely" distance for GRS 1758-258 of ~ 8.5 kpc that would be inferred from the observations already carried out with GTC and HST-NICMOS.

I also suggest the authors continue this line of research by a search on the molecular cloud for indications of jet impacts by high resolution observations at millimeter wavelengths.

Reviewer #2 (Remarks to the Author):

Report on "A Galactic microquasar mimicking winged radio galaxies"

The authors report on a deep radio imaging study of the known microquasar source GSR 1758-258. By stacking new and archival VLA data with sensitivity to a range of spatial scales, they detect large-scale, diffuse emission asymmetrically located with respect to the bright radio sources to the north and south of the microquasar, which they interpret as back-flow from hot spots. They draw a parallel between the appearance of this source and the morphologies of winged radio galaxies. The brightness and length asymmetries between northern and southern lobes are interpreted as due to the northern jet running into a molecular cloud, which has given rise to a density gradient between the two lobes, and hence affects their morphologies.

This is a nicely-presented, well-argued case study of a well-known microquasar source. I find the authors' interpretation of the data reasonably compelling. I agree that the precession hypothesis seems unlikely, and the black hole merger scenario even less likely. So the interpretation of the emission as hotspots and backflow into the lobes, with the asymmetry in brightness and length being due to a density gradient, seems to be a plausible explanation for the observed structure.

What I find to be missing in this paper is the key insight that would appeal to a broad community outside the X-ray binary field. The authors mention in the first paragraph that lessons learned from this case will open a new perspective in the analogy between microquasars and their more massive counterparts in AGN. In the second paragraph of the main text, they suggest that similarities between radio galaxies and microquasars could help put models for WRGs to the test. However, having established the most plausible explanation for the appearance of GRS 1758-258, they do not then attempt to explain what consequences this has for our understanding of WRGs. The only comments they provide on this topic towards the end of the paper are a suggestion that further modelling of this system would enable further studies of the scaled-down WRG phenomenon, and that additional WRG-like microquasars may be detected in future using more sensitive radio interferometers. Without being able to provide some key implications for the nature of radio galaxies, I do not feel that this paper would have a sufficiently broad appeal to merit publication in Nature Communications. In its current form, it would be more appropriate for a more specialist astrophysics journal.

My other comment is that the authors rely only on the observed morphology of the source and the astrometry of the northern hotspot to inform their conclusions. Given the large bandwidth available in the 2016 observations, I wonder whether the authors might be able to provide some spectral information to back up their conclusions; do the observed source spectra make sense in

light of the preferred interpretation? Equally, there could be some higher/lower-frequency data in the archives that might aid a spectral comparison (although sensitivity is clearly a key issue in detecting the lobe structures shown in Figure 1).

In what follows, I provide some more detailed comments on specific parts of the manuscript.

Main text:

- In the first paragraph, I would like to see the authors quantify number/fraction of low-luminosity radio galaxies exhibiting the kinds of extended lobes that give rise to the WRG classification. How common are WRGs? The generalist reader could not be expected to know this. I would also like to see some additional references in the first sentence or two of this paragraph.
- In paragraph 3, the authors mention that they selected certain VLA archival radio observations. I would like to see the Methods section provide more detail on the exact selection criteria used, beyond the general requirement for the 6cm band and C configuration, and a generic mention of the "best" datasets in D and DnC configurations. What denotes "best"? A minimum exposure time or elevation? Please clarify.
- In paragraph 4, the authors constrain a precession cone opening angle to be at most a couple of degrees. With only three points, I worry as to how well any precession period can actually be sampled, so view this statement as somewhat speculative.
- At the end of paragraph 4, the authors mention a "very specific modification" in the process of star mass loss. I would like to know more about what that modification would need to be, and how likely or unlikely that is. Also, in the previous sentence, "de" should read "the".
- The CO gas cloud shown in Fig 3 is centred significantly (~ 45 arcmin) to the N of the target. The image in Fig 3 is much larger than that in Fig 1. The northern hotspot is located ~ 1 arcmin from the core, and yet the scale bar of 15 arcmin in Fig 3 suggests that the hotspot is likely to be located somewhat south of the bottom contour of the molecular cloud. That said, the cloud shown does not end abruptly, so its outer regions are likely to extend down towards GRS 1758-258. But it would be useful for the authors to make that assumption clear. Specifically, the statement in paragraph 6 that the CO brightness profile would be consistent with the cloud density changing by a factor of 4 on a linear size scale comparable to that of the extended jets is surprising given that the size scale of the extended jets in Fig 1 is smaller than a resolution element in Fig 3. Some more details would be useful to back up this assertion.

Methods:

- The authors mention that both AIPS and CASA were used for data reduction. I would like to know whether the authors have done any consistency checks to see whether the specific package used introduces any systematic shift in the results (especially between the old VLA data and the newer EVLA data).
- I would like to know whether the authors checked that the assumed position for the phase calibrator source J1751-253 was the same for both archival and new data sets, as shifts in the assumed position could affect the astrometry presented in Figure 2. While I would be very surprised if the position shifted by as much as 2 arcseconds, I think it would be an important check to do, since the shift in the measured position of the northern hotspot is a key plank of the interpretation.
- On the topic of the northern hotspot position, I note that the northern hotspot appears to be extended in the map presented in Figure 1, whereas the core appears to be unresolved. Given the different uv-coverage of the epochs in 1997, 2008 and 2016, I would like the authors to check whether the different sampling of the resolved structure could have contributed to the plotted

positional shifts in Figure 2. For instance, using the deepest image (shown in Figure 6) as a model, that model could be used to generate a simulated data set using the uv-coverage of the 1997 and 2008 observations, and the positions remeasured. As previously stated, ruling out any possible systematic effects in the positional shifts measured in Figure 2 would help establish the veracity of the authors' interpretation for the observed morphology.

- The authors should state whether any reweighting of any of the data sets was used in DBCON, as is often applied when combining data from different array configurations.

- The authors might wish to quantify the degree of bandwidth smearing affecting the target at the position of the northern hotspot (furthest from the core), to demonstrate that this does not markedly affect the observed morphology.

Figures:

- In the caption to Figure 1, the authors state that sources #1, #2 and #3 are unrelated. While Marti et al. (2015) also make a similar statement, and Marti et al. (2002) say that source #2 (their VLA-D) is unrelated although clearly non-thermal, it is not clear to me what evidence forms the basis for these claims, other than the probabilistic calculation in Marti et al. (2002) that there is a 10% probability of finding a 0.14-mJy source in a 1 square arcminute region. It might be helpful for the authors to make either a probability-based argument or to provide other evidence for the fact that these other sources are unrelated.

- In the caption to Figure 4, the authors reference paper 27 (Kaiser & Alexander 1997), which does not appear to have a Milky Way rotation curve in it. I suggest that the references be checked thoroughly.

- The caption to Figure 7 would benefit from some additional explanation. The choice of contouring level seems a little arbitrary; it is neither a strict arithmetic progression, nor a strict geometric progression. A geometric progression is typically used in radio astronomy to avoid overstating the significance of low signal-to-noise features, but I do not mandate that. Rather, I would simply request a logical, simple contouring scheme that does not require fine-tuning to emphasise or hide certain features of the image.

Tables:

- Table 1 would benefit from listing the rms noise levels in each of the individual images considered, to show their relative sensitivities, and hence which observations were likely to dominate the stacked data set.

Reviewer #3 (Remarks to the Author):

The authors present a deep radio image of the microquasar GRS 1758-258. This image shows the presence of extended emission originating from the two hot spots producing an overall Z-shape morphology, somewhat similar to that seen in winged radiogalaxies. They convincingly argue that this is due to jet material flowing back from the jet termination points towards the nucleus and ascribe the observed asymmetry to the differences in the ISM density in the EW direction.

Backflows are naturally produced whenever a jet has a density smaller than the surrounding medium. I do not find that the finding that a microquasar launches a low density jet sheds a novel light on the properties of these objects. The similarity with winged RGs is also expected in the backflow scenario and it does not represent a significant step forward in the connection between galactic and stellar black holes.

Reply to all reviewer's comments (in blue):

Reviewers' comments: Reviewer #1 (Remarks to the Author):

Based on the microquasar-quasar analogy it is proposed a new way to gain insight on the physics mechanisms that produce the morphology of winged radio galaxies. The reported observations of GRS 1758-258 clearly show a Z-shaped morphology similar to that found in radio galaxies, and because of the relative short time scales in the microquasar, the authors can conclude that the Z-shaped morphology in this source is produced by hydrodynamical backflows of jet shocked material propagating in an inhomogeneous environment. The subject and reported observations are of topical interest and I strongly recommend publication in Nature.

I suggest the authors briefly discuss the "likely" distance for GRS 1758-258 of ~ 8.5 kpc that would be inferred from the observations already carried out with GTC and HST-NICMOS. I also suggest the authors continue this line of research by a search on the molecular cloud for indications of jet impacts by high resolution observations at millimeter wavelengths.

We have added some sentences justifying the distance value adopted as supported by spectral energy distribution fits based on GTC and HST data, as well as compliance with the causality upper limit from structural variability in the northern hot spot. We agree with the referee about the need of CO observations with improved angular resolution in addition to other density tracers. This should be carried out in future work.

Reviewer #2 (Remarks to the Author): Report on "A Galactic microquasar mimicking winged radio galaxies"

The authors report on a deep radio imaging study of the known microquasar source GSR 1758-258. By stacking new and archival VLA data with sensitivity to a range of spatial scales, they detect large-scale, diffuse emission asymmetrically located with respect to the bright radio sources to the north and south of the microquasar, which they interpret as back-flow from hot spots. They draw a parallel between the appearance of this source and the morphologies of winged radio galaxies. The brightness and length asymmetries between northern and southern lobes are interpreted as due to the northern jet running into a molecular cloud, which has given rise to a density gradient between the two lobes, and hence affects their morphologies.

This is a nicely-presented, well-argued case study of a well-known microquasar source. I find the authors' interpretation of the data reasonably compelling. I agree that the precession hypothesis seems unlikely, and the black hole merger scenario even less likely. So the interpretation of the emission as hotspots and backflow into the lobes, with the asymmetry in brightness and length being due to a density gradient, seems to be a plausible explanation for the observed structure.

What I find to be missing in this paper is the key insight that would appeal to a broad community outside the X-ray binary field. The authors mention in the first paragraph

that lessons learned from this case will open a new perspective in the analogy between microquasars and their more massive counterparts in AGN. In the second paragraph of the main text, they suggest that similarities between radio galaxies and microquasars could help put models for WRGs to the test. However, having established the most plausible explanation for the appearance of GRS 1758-258, they do not then attempt to explain what consequences this has for our understanding of WRGs. The only comments they provide on this topic towards the end of the paper are a suggestion that further modelling of this system would enable further studies of the scaled-down WRG phenomenon, and that additional WRG-like microquasars may be detected in future using more sensitive radio interferometers. Without being able to provide some key implications for the nature of radio galaxies, I do not feel that this paper would have a sufficiently broad appeal to merit publication in Nature Communications. In its current form, it would be more appropriate for a more specialist astrophysics journal.

To address this referee comment, certainly the most relevant one, we have rewritten several sections of the paper while adapting it to the Nature Comm. format. Now, we stress the winged radio galaxy problem and how the GRS 1758-258 helps to narrow the physical scenarios behind it by strengthening the plausibility argument of the backflow model. We consider that our first detection of a Z-type morphology in a microquasar offers a testbench where radio galaxy models can be compared to a nearby and fast-evolving system. In our view, this should be considered as a matter of general interest for both the galactic and extragalactic astrophysical communities. In particular, supporting the case for a backflow interpretation in WRGs has also consequences on the other scenarios discussed, as e.g. in the rate of spin-flip events in supermassive black holes and its connection with the gravitational wave background (although the study of these consequences is beyond the scope of our work).

My other comment is that the authors rely only on the observed morphology of the source and the astrometry of the northern hotspot to inform their conclusions. Given the large bandwidth available in the 2016 observations, I wonder whether the authors might be able to provide some spectral information to back up their conclusions; do the observed source spectra make sense in light of the preferred interpretation? Equally, there could be some higher/lower- frequency data in the archives that might aid a spectral comparison (although sensitivity is clearly a key issue in detecting the lobe structures shown in Figure 1).

The Z-shaped extensions are a reliable detection with a convincing signal-to-noise ratio $SNR \sim 6$. Unfortunately, this is not as good as one would desire ($SNR > 10$). Splitting the data into two frequency sets renders the spectral index determination very uncertain for this purpose. On the other hand, the amount of data in the archives that could provide a matching-beam data set at L or X-band is not enough to produce a good map with the required sensitivity at the few microJy level.

In what follows, I provide some more detailed comments on specific parts of the manuscript. Main text:

- In the first paragraph, I would like to see the authors quantify number/fraction of

low-luminosity radio galaxies exhibiting the kinds of extended lobes that give rise to the WRG classification. How common are WRGs? The generalist reader could not be expected to know this. I would also like to see some additional references in the first sentence or two of this paragraph.

The paper includes now sentences and references about the frequency of occurrence for WRGs.

- In paragraph 3, the authors mention that they selected certain VLA archival radio observations. I would like to see the Methods section provide more detail on the exact selection criteria used, beyond the general requirement for the 6cm band and C configuration, and a generic mention of the "best" datasets in D and DnC configurations. What denotes "best"? A minimum exposure time or elevation? Please clarify.

The selection criteria are clarified in the Methods section and they are mainly based on a minimum exposure time. By the way, when checking this item we noticed that a suitable VLA observing run in 1992 September 10 was omitted from the analysis. We have reprocessed and re-measured everything in order include it and slightly improve the results. This is why some of the maps, figures and tables are slightly different than the previous version of the paper.

- In paragraph 4, the authors constrain a precession cone opening angle to be at most a couple of degrees. With only three points, I worry as to how well any precession period can actually be sampled, so view this statement as somewhat speculative.

We have relaxed the statement from a couple of degrees to a few degrees and clarified the statement stating that significant changes in the position angle of the GRS 1758-258 jets have never been reported. No mention to a possible precession period is present.

- At the end of paragraph 4, the authors mention a "very specific modification" in the process of star mass loss. I would like to know more about what that modification would need to be, and how likely or unlikely that is.

We have rewritten the end of paragraph 4 to account for research on jet formation that was not considered in the previous version of the manuscript. Still, the need of a strong and specific change in the accreted matter on time scales of many years is required. We prefer not to go into details on this though, as the discussion would be very speculative and may unbalance the text, but we indicate now that a main sequence star would be at odds with such a significant change in the mass-transfer to the compact object.

Also, in the previous sentence, "de" should read "the".

Done.

- The CO gas cloud shown in Fig 3 is centred significantly (~45 arcmin) to the N of the target. The image in Fig 3 is much larger than that in Fig 1. The northern hotspot

is located ~ 1 arcmin from the core, and yet the scale bar of 15 arcmin in Fig 3 suggests that the hotspot is likely to be located somewhat south of the bottom contour of the molecular cloud. That said, the cloud shown does not end abruptly, so its outer regions are likely to extend down towards GRS 1758-258. But it would be useful for the authors to make that assumption clear. Specifically, the statement in paragraph 6 that the CO brightness profile would be consistent with the cloud density changing by a factor of 4 on a linear size scale comparable to that of the extended jets is surprising given that the size scale of the extended jets in Fig 1 is smaller than a resolution element in Fig 3. Some more details would be useful to back up this assertion.

We have clarified this point by including the new Figure 4 showing a slice of CO emission. By fitting a parabola to the Dame data, the 4x variation on a scale of a few times the jet length appears plausible.

Methods:- The authors mention that both AIPS and CASA were used for data reduction. I would like to know whether the authors have done any consistency checks to see whether the specific package used introduces any systematic shift in the results (especially between the old VLA data and the newer EVLA data).

Following this comment, we recomputed AIPS and CASA maps and compared the positions of point sources in the field. Offsets turned out to be on average less than $1/60$ of the synthesized beam in the direction with less angular resolution (North-South). Some sentences about this have been added to the Methods section. In any case, mixing of mapping tasks was avoided and all final imaging was carried out using AIPS task IMAGR for all data sets. This is now clearly stated in the Methods text.

- I would like to know whether the authors checked that the assumed position for the phase calibrator source J1751- 253 was the same for both archival and new data sets, as shifts in the assumed position could affect the astrometry presented in Figure 2. While I would be very surprised if the position shifted by as much as 2 arcseconds, I think it would be an important check to do, since the shift in the measured position of the northern hotspot is a key plank of the interpretation.

Yes, this is a good point! The calibrator position was improved and a slightly different value was adopted for 2008 and 2016 data as compared to 1992 and 1997. We checked and the shift is -0.0021 s in R.A. and $-0.257''$ in DEC. Figure 2 and Table 2 have been corrected for this effect. A plotting error in the 2008 point in Fig. 2 has been also corrected (this did not affect the table). Nevertheless, all these changes leave untouched the conclusions of the paper.

- On the topic of the northern hotspot position, I note that the northern hotspot appears to be extended in the map presented in Figure 1, whereas the core appears to be unresolved. Given the different uv-coverage of the epochs in 1997, 2008 and 2016, I would like the authors to check whether the different sampling of the resolved structure could have contributed to the plotted positional shifts in Figure 2. For instance, using the deepest image (shown in Figure 6) as a model, that model could be used to generate a simulated data set using the uv-coverage of the 1997 and

2008 observations, and the positions remeasured. As previously stated, ruling out any possible systematic effects in the positional shifts measured in Figure 2 would help establish the veracity of the authors' interpretation for the observed morphology.

We carried out this test as proposed. The clean components of the 2016 epoch were adopted as a model and sampled with the uv -coverage of 1997 and 2008. Then, we re-imaged the resulting visibility data. Ideally, one would expect the core and hot spot positions to remain the same as in 2016.

Figure 1. Testing the effect of Fourier plane sampling using the 2016 map as a model. Scale is the same as in Fig. 2 of the paper for easy comparison.

In practice, the result is the one shown in Fig. 1 above. The core position is recovered in all cases within 0.2 arc-second. The same occurs for the northern hot spot, although for this extended feature the consistency is within 0.5 arc-second for the 2008 uv coverage. In any case, the position offsets displayed in the paper's Fig. 2 (> 2 arc-second) are significantly above the possible magnitude of this systematic effect. This is now commented in the Methods section. The Fig. 1 of this reply is **not included in the paper**, but shown here only for the referees. In case any of you wish it to be included, we will be glad to do so.

- The authors should state whether any reweighting of any of the data sets was used in DBCON, as is often applied when combining data from different array configurations.

When combined data sets from different project codes in Table 1, we adopted a re-weighting criterion based on the sum of gridding weights as reported by the AIPS task IMAGR. The AIPS re-weighting parameters were adjusted in such a way that each project code was weighted according to the inverse squared of its rms noise, as listed in Table 1. The last column of the Table gives the final relative weight of each project in its contribution to the final image in Fig. 1. This is now explained in the Methods section.

- The authors might wish to quantify the degree of bandwidth smearing affecting the target at the position of the northern hotspot (furthest from the core), to demonstrate that this does not markedly affect the observed morphology.

Positions of the Northern hot spot in 1997 and 2008 were measured from their corresponding individual maps. The relatively narrow bandwidth available at that epoch renders chromatic aberration problems not an issue. About the 2016 data, we checked that the Table 1 position is consistent with 2016 CASA maps computed using the clean multi-frequency synthesis mode, with negligible smearing effects.

Bandwidth smearing could play a role when DBCON-combining historical and modern VLA data acquired with noticeably different frequencies, even if all of them are within the C-band. We have tried to approximately estimate its effects in Fig. 1 map, and they could degrade the beam and sensitivity at the 20-25% level. This is now mentioned in the Methods section. One could fear this casts some doubts on the reality of backflow emission. However, we are confident that this is not the case because the comparison of the AIPS map in Fig. 1 map with the CASA map Supplementary Fig. 4 (free from smearing) is very reassuring. Despite their different sensitivity to extended emission, the same backflow structures can be well traced in both of them.

Figures:

- In the caption to Figure 1, the authors state that sources #1, #2 and #3 are unrelated. While Marti et al. (2015) also make a similar statement, and Marti et al. (2002) say that source #2 (their VLA-D) is unrelated although clearly non-thermal, it is not clear to me what evidence forms the basis for these claims, other than the probabilistic calculation in Marti et al. (2002) that there is a 10% probability of finding a 0.14-mJy source in a 1 square arcminute region. It might be helpful for the authors to make either a probability-based argument or to provide other evidence for the fact that these other sources are unrelated.

The sources #1, #2 and #3 are proposed as unrelated objects because of their compact nature. They remain detected even in super-resolution maps with pure uniform weighting. In the same maps, the rest of jet extended emission vanishes and even the northern hot spot is hard to see. It does not seem plausible that the jets of the microquasar have so many compact components scattered in the field of view. This argument is now mentioned when introducing Fig. 1.

- In the caption to Figure 4, the authors reference paper 27 (Kaiser & Alexander 1997), which does not appear to have a Milky Way rotation curve in it. I suggest that the references be checked thoroughly.

Yes. It was a mistake. The references have been corrected and checked. New references have been also added while rewriting the paper to emphasize its general interest.

- The caption to Figure 7 would benefit from some additional explanation. The choice of contouring level seems a little arbitrary; it is neither a strict arithmetic progression,

nor a strict geometric progression. A geometric progression is typically used in radio astronomy to avoid overstating the significance of low signal-to-noise features, but I do not mandate that. Rather, I would simply request a logical, simple contouring scheme that does not require fine-tuning to emphasise or hide certain features of the image.

We have adopted a counteracting scheme based on a geometric progression starting from a conservative 4σ level. The new contour map is now in Supplementary Fig. 1.

Tables:

- Table 1 would benefit from listing the rms noise levels in each of the individual images considered, to show their relative sensitivities, and hence which observations were likely to dominate the stacked data set.

This has been done in the context of clarifying the weighting scheme used with the DBCON task.

Reviewer #3 (Remarks to the Author):

The authors present a deep radio image of the microquasar GRS 1758-258. This image shows the presence of extended emission originating from the two hot spots producing an overall Z-shape morphology, somewhat similar to that seen in winged radiogalaxies. They convincingly argue that this is due to jet material flowing back from the jet termination points towards the nucleus and ascribe the observed asymmetry to the differences in the ISM density in the EW direction.

Backflows are naturally produced whenever a jet has a density smaller than the surrounding medium. I do not find that the finding that a microquasar launches a low density jet sheds a novel light on the properties of these objects. The similarity with winged RGs is also expected in the backflow scenario and it does not represent a significant step forward in the connection between galactic and stellar black holes.

Here we must comment that one thing is that a phenomenon is expected to occur according to our current theoretical understanding of jets, while something very different is that the phenomenon is actually observed to occur. To our knowledge, this is the first time that a backflow is detected in a microquasar. If this is not a significant step forward, certainly it is paving the way for it by allowing theorists to model the backflow phenomenon in a completely new physical context.

Reviewers' comments:

Reviewer #1 (Remarks to the Author):

Do not incorporate in the manuscript full responses to referee #2, make a synthesis of them. Mention the possibility of observing in microquasars the evolution of jets at large distance scales from their sources, and done in 1E 1740, and in the future, backflows as in GRS 1758.

Reviewer #2 (Remarks to the Author):

I thank the authors for their comprehensive response to my original comments. I find that they have addressed the majority of the points satisfactorily, although I have a few remaining questions, suggestions and comments, as laid out below.

My main topic of concern had been whether this work would have broad implications and appeal to a community beyond those interested only in Galactic microquasars. The authors have provided much more background to the phenomenon of WRGs in the Introduction section, which helps set up the problem being addressed. They have also spent the final few paragraphs of the Discussion section providing the broader context of their findings. If they are correct that their findings can be so easily generalised from microquasars to WRGs, then this result would indeed have the broader implications that would argue for its publication in Nature Communications. However, to my mind the authors somewhat overstate the significance of their result. They argue that finding a well-established hydrodynamical backflow in a single microquasar enables them to exclude all other scenarios in the general case of WRGs, which I find to be a logical step too far. The environments of radio galaxies and those of microquasars can differ significantly (see, e.g., discussion in Heinz 2002, A&A, 388, L40), so to explain all observed WRGs (X/Z-shaped radio galaxies) with the same scenario as seen in a single case of a different class of objects seems to me to be a little bit of a stretch. I would accept the claim that this firmly establishes the plausibility of a hydrodynamical backflow being able to give rise to a Z-shaped morphology, but not that this then has to hold for the entire class of WRG sources. For the same reason, I am not persuaded by the claim that spin-flip events are therefore significantly less frequent in the WRG population (a statement which in any case would need to be quantified if it were to be included). I strongly recommend that the authors reword this discussion, making it clear which of their statements are robust, and which are of a more speculative nature.

In addition to that major concern, I had a few more minor comments:

- In the third paragraph of the Introduction, the authors mention with the strong parallels between microquasars and radio galaxies/AGN. The mass scaling they invoke is indeed well-established close to the compact object, where the black hole dominates. However, on large scales where the lobes form, the evolution is also affected by the relative powers of the jets and the relative densities of the surrounding media (see, e.g., Heinz 2002), so the parallels may not be as clear in many cases.

- In the first paragraph of the Discussion section, where the jet realignment time is discussed, I would recommend citing Maccarone (2002, MNRAS, 336, 1371), which both predates the current reference to Machalski et al. (2016) and is specifically applied to microquasars. At the end of this paragraph, I would recommend quoting the inferred jet age for GRS 1758-258, to save the reader having to scan forward several pages to make the comparison.

- On p8, "blackflow" -> "backflow"

- Same page: The authors state that the positive/negative E-W density gradients required to provide the asymmetric backflow morphology with respect to the jet axis are compatible with the CO distribution shown in Fig 3. While I agree that Fig 3 argues for a N-S density gradient, I don't

think it can say anything about E-W gradients, so suggest that the authors rephrase this statement to be clearer as to what is meant.

- Same page, and Fig 4: The authors mention a parabolic fit to the profile of the CO emission. I don't understand the logic for fitting a parabolic profile. I see no reason why the CO brightness should rise again moving to the south of the microquasar. I would have thought that a power law, exponential or Gaussian wing giving a uniform decrease in CO brightness on moving from north to south would have been more appropriate here.

- In the final paragraph of the Discussion section, the authors claim that additional WRG-like microquasars will routinely emerge when more sensitive interferometers are available. I note that in the authors' interpretation, the observed morphology owes much to the density enhancement due to the nearby molecular cloud. If the authors are to claim that this phenomenon will become more common, they need to establish the likelihood of a microquasar jet running into a molecular cloud (perhaps taking into account the number density of both classes of object).

- On a similar note, I would class a number of existing arrays, such as the VLA, MeerKAT and eMERLIN as modern interferometers, and not only the SKA.

- Methods: What drives the requirements for observations to have been accumulated within a week? Is the source believed to change on short timescales? If longer-timescale variations are believed to affect the image stacking, then would they not also affect the stack of images from 1992-2016?

- In the bandwidth smearing discussion, the authors mention a fractional bandwidth of $\sim 1/5$, but the data are dominated by the 2016 data, which has 2 GHz of bandwidth centered at 5.5 GHz, implying a fractional bandwidth of $< 1/3$.

- On the last page of the methods section, the authors reference Fig 7, which I think should be Extended Data/Supplementary Fig 4.

- Table 1: Were all the CD/D observations stacked together before adding to other configurations via DBCON? I don't understand the difference between horizontal lines and + signs in the table. Please clarify the procedure.

- Supplementary Data Fig 4: Please state explicitly the units of the color bar. I assume Jy/beam?

Reviewer #3 (Remarks to the Author):

The authors addressed the issue I raised only very briefly. I remain unconvinced that these results represent a significant step forward in our understanding of the jets phenomenon.

Reviewers' comments:

Reviewer #1 (Remarks to the Author):

Do not incorporate in the manuscript full responses to referee #2, make a synthesis of them. Mention the possibility of observing in microquasars the evolution of jets at large distance scales from their sources, and done in 1E 1740, and in the future, backflows as in GRS 1758.

The possibility of studying the almost real-time evolution of microquasar jets at long distances from the central source is now mentioned in our conclusive paragraph.

We also thank this referee for his advice on how to address referee #2 comments. Indeed, we have tried to be as synthetic as possible in order not to increase the paper length beyond the necessary.

Concerning the 1E 1740.7-2942 case, we already carried out Jansky VLA observations of it but the data is still under analysis. Here, the detection of winged features turns out to be harder than in GRS 1758 because of severe confusion with extended emission from the Galactic Center, that is very close for this source.

Reviewer #2 (Remarks to the Author):

I thank the authors for their comprehensive response to my original comments. I find that they have addressed the majority of the points satisfactorily, although I have a few remaining questions, suggestions and comments, as laid out below.

My main topic of concern had been whether this work would have broad implications and appeal to a community beyond those interested only in Galactic microquasars. The authors have provided much more background to the phenomenon of WRGs in the Introduction section, which helps set up the problem being addressed. They have also spent the final few paragraphs of the Discussion section providing the broader context of their findings. If they are correct that their findings can be so easily generalised from microquasars to WRGs, then this result would indeed have the broader implications that would argue for its publication in Nature Communications. However, to my mind the authors somewhat overstate the significance of their result. They argue that finding a well-established hydrodynamical backflow in a single microquasar enables them to exclude all other scenarios in the general case of WRGs, which I find to be a logical step too far. The environments of radio galaxies and those of microquasars can differ significantly (see, e.g., discussion in Heinz 2002, A&A, 388, L40), so to explain all observed WRGs (X/Z-shaped radio galaxies) with the same scenario as seen in a single case of a different class of objects seems to me to be a little bit of a stretch. I would accept the claim that this firmly establishes the plausibility of a hydrodynamical backflow being able to give rise to a Z-shaped morphology, but not that this then has to hold for the entire class of WRG sources. For the same reason, I am not persuaded by the claim that spin-flip events are therefore significantly less frequent in the WRG population (a statement which in any case would need to be quantified if it were to be included). I

strongly recommend that the authors reword this discussion, making it clear which of their statements are robust, and which are of a more speculative nature.

We are very grateful to referee #2 for his/her valuable comments that are really helping to improve the manuscript. To address this main concern, we have reworded the key paragraphs of the discussion. First, we neatly separate our robust finding of a Z-morphology in GRS 1758-258 with a likely hydrodynamic origin from our more speculative implications. Secondly, following the referee reasonable views, we are happy to moderate our statement about the exclusion of other scenarios. In particular, we clearly mention that the proposed hydrodynamic origin for WRGs, although strongly supported by our results, does not fully rule out other alternatives to take place in the Universe. Moreover, the abstract has been also modified to reflect this more cautious view.

In addition to that major concern, I had a few more minor comments: □- In the third paragraph of the Introduction, the authors mention with the strong parallels between microquasars and radio galaxies/AGN. The mass scaling they invoke is indeed well-established close to the compact object, where the black hole dominates. However, on large scales where the lobes form, the evolution is also affected by the relative powers of the jets and the relative densities of the surrounding media (see, e.g., Heinz 2002), so the parallels may not be as clear in many cases.

Yes, this parallelism may not be as clear owing to the very different environments and jet powers and, therefore, hard to be observationally confirmed. Nevertheless, dynamic similarity is an extremely robust tool of fluid mechanics and, based on it, we expect the same underlying laws to rule the behavior of hydrodynamic jets no matter the scale. At the end of this 3rd paragraph, we have now added a sentence expressing this as an a priori expectation that will appear to be later supported by our GRS 1758-258 finding.

- In the first paragraph of the Discussion section, where the jet realignment time is discussed, I would recommend citing Maccarone (2002, MNRAS, 336, 1371), which both predates the current reference to Machalski et al. (2016) and is specifically applied to microquasars. At the end of this paragraph, I would recommend quoting the inferred jet age for GRS 1758-258, to save the reader having to scan forward several pages to make the comparison.

Done. Affected references have been also renumbered and checked again.

- On p8, "blackflow" -> "backflow"

Typo corrected.

- Same page: The authors state that the positive/negative E-W density gradients required to provide the asymmetric backflow morphology with respect to the jet axis are compatible with the CO distribution shown in Fig 3. While I agree that Fig 3 argues for a N-S density gradient, I don't think it can say anything about E-W gradients, so suggest that the authors rephrase this statement to be clearer as to what is meant.

Ok. We have rephrased this part of the text to make clear that Fig. 3 cannot confirm, nor rule out, the existence of a possible E-W gradient. Only N-S gradient is obvious.

- Same page, and Fig 4: The authors mention a parabolic fit to the profile of the CO emission. I don't understand the logic for fitting a parabolic profile. I see no reason why the CO brightness should rise again moving to the south of the microquasar. I would have thought that a power law, exponential or Gaussian wing giving a uniform decrease in CO brightness on moving from north to south would have been more appropriate here.

We used a second order polynomial or parabolic function as one of the simplest way to represent the antenna temperature data varying over a finite interval. There was no a priori assumption on the growth, or decrease, of this variable. Anyway, we have changed the fit and used an exponential function instead. Our conclusions are not affected and the text remains the same except when quoting the type of fit.

- In the final paragraph of the Discussion section, the authors claim that additional WRG-like microquasars will routinely emerge when more sensitive interferometers are available. I note that in the authors' interpretation, the observed morphology owes much to the density enhancement due to the nearby molecular cloud. If the authors are to claim that this phenomenon will become more common, they need to establish the likelihood of a microquasar jet running into a molecular cloud (perhaps taking into account the number density of both classes of object).

While the number of star-forming regions in the Galaxy is relatively well established, the census of microquasars remains still very uncertain for a reliable likelihood estimate. Therefore, we have approached this comment by checking how many microquasars are currently known with their ejecta having some kind of reported interaction or connection with ISM clouds. In addition to GRS 1758-258 and the Cygnus X-1 jet-driven bubble, this includes at least XTE J1748-288, XTE J1550-564, 1E1740.9-2942, and GRS 1915+105. The fact that this approaches about one third of the whole Galactic microquasar population encourages us to consider the possibility of future wing structures being detected when micro-Jy sensitivities become routinely achievable. Sentences and new references about this are now included in the text. At the same time, we have moderated our former optimism about this expectation.

- On a similar note, I would class a number of existing arrays, such as the VLA, MeerKAT and eMERLIN as modern interferometers, and not only the SKA.

Future sensitive interferometers, in addition to the existing VLA, are now being mentioned.

- Methods: What drives the requirements for observations to have been accumulated within a week? Is the source believed to change on short timescales? If longer-timescale variations are believed to affect the image stacking, then would they not also affect the stack of images from 1992-2016?

There is no variability issue regarding the week criterion. It was just to simplify and homogenize the selection of VLA data by giving preference to historical VLA runs closely packed in time for easier retrieval and calibration. Of course, a few potentially usable data sets were excluded in this process. However, these mainly consisted of snapshots for long-term monitoring purposes. We do not consider that this minor shortcoming has a significant impact on our current results beyond marginal effects. As explained in the Methods section, the only variability issue taken into account was to exclude some VLA runs where the central core of GRS 1758-258 was clearly undetected.

- In the bandwidth smearing discussion, the authors mention a fractional bandwidth of $\sim 1/5$, but the data are dominated by the 2016 data, which has 2 GHz of bandwidth centered at 5.5 GHz, implying a fractional bandwidth of $< 1/3$.

The fractional bandwidth value mentioned in the paper was indeed a very crude one and the referee is right that we should consider a slightly higher value. However, accurately estimating the smearing effects in our case is by no means a straightforward task. This is because we are combining data with very different fractional bandwidths, while most related textbooks and interferometer manuals always assume a single one. In order to circumvent this issue, we re-did the simple smearing estimates in the paper by adopting the fractional bandwidth value averaged over the whole data sets. Certainly, here the 2016 data dominates at the 75% level. From Table 1, the resulting value is close to $1/4$ in agreement with the $< 1/3$ value quoted by the referee. This part of the text has been re-written and also a new reference is being quoted instead of a link to the VLA web page. In any case, we stress again here that the comparison with Fig. 1 with Suppl. Fig. 4 reinforces the reality of the backflow structures against any smearing criticism.

- On the last page of the methods section, the authors reference Fig 7, which I think should be Extended Data/Supplementary Fig 4.

The referee is right. We have updated the reference to this figure that still used the old numbering scheme of the paper.

- Table 1: Were all the CD/D observations stacked together before adding to other configurations via DBCON? I don't understand the difference between horizontal lines and + signs in the table. Please clarify the procedure.

Yes, CD and D configuration projects were separately stacked first before merging with the combined C configuration runs. This was the idea behind the + signs. But it is true that this may cause confusion and we simply removed the +'s from Table 1. Clarifying sentences about this have been also added when describing the merging of VLA data sets and as a footnote in Table 1.

- Supplementary Data Fig 4: Please state explicitly the units of the color bar. I assume

Jy/beam?

Yes, these units are correct. We have added them together with a description of the synthesized beam that was also missing.

Reviewer #3 (Remarks to the Author):

The authors addressed the issue I raised only very briefly. I remain unconvinced that these results represent a significant step forward in our understanding of the jets phenomenon.

The issue raised by this referee in his/her previous comments consisted only of a strong negative opinion about our results representing a significant step forward in the connection between galactic and stellar black holes. Otherwise, the comments did not include any objection about the reliability of our backflow detection nor its physical interpretation as of hydrodynamic origin. In this context, it is very difficult for the authors to address the referee concerns as they were expressed in a very general way.

Finally, we stress again the fact that this is the first reported detection of backflow in a microquasar.

REVIEWERS' COMMENTS:

Reviewer #2 (Remarks to the Author):

I thank the authors for responding to my previous comments so thoroughly. They have toned down the language appropriately in most cases, and I appreciate the separation of the robust results from the more speculative inferences.

With this separation, and the updated, more appropriate discussion of the implications, I feel that this paper could potentially be appropriate for publication in Nature Communications. The work does provide new insights into possible scenarios for winged radio galaxies, showing that the hydrodynamical backflow scenario is plausible. Drawing parallels between AGN and their stellar-mass analogues has been a goal of the microquasar field since its inception, and this is a striking example of such a parallel.

I have a few minor comments remaining that I would want the authors to address before acceptance:

- Line 31: I find that "infer this mechanism to be at work in many extragalactic cases" is a little too strong. Better would be "suggest that this mechanism could also be at work" or "hypothesise that this mechanism could also be at work". This is not an inference, but a hypothesis/speculation.

- Lines 73-78: As mentioned before, I would like to see a reference to Heinz (2002) in this discussion - either here, or possibly on line 236.

- Line 133: The jet alignment timescale estimates assume a constant mass accretion rate and disc alpha viscosity parameter. Please mention this assumption explicitly, as most black hole X-ray binary systems are transient (see Coriat et al. 2012, who note explicitly that the persistent nature of GRS1758-258 is unclear). Should the source be a long duty cycle transient, the alignment timescale would be longer.

- Line 248: "yields another remarkable implication" is again a little too strong, since it is followed by a reasonable remark about speculating further on the inferences. I would recommend toning that statement down.

- Line 257: eMERLIN is a current instrument that is already available, and has been for several years.

Reviewers' comments:

Reviewer #2 (Remarks to the Author):

I thank the authors for responding to my previous comments so thoroughly. They have toned down the language appropriately in most cases, and I appreciate the separation of the robust results from the more speculative inferences.

With this separation, and the updated, more appropriate discussion of the implications, I feel that this paper could potentially be appropriate for publication in Nature Communications. The work does provide new insights into possible scenarios for winged radio galaxies, showing that the hydrodynamical backflow scenario is plausible. Drawing parallels between AGN and their stellar-mass analogues has been a goal of the microquasar field since its inception, and this is a striking example of such a parallel.

I have a few minor comments remaining that I would want the authors to address before acceptance:

- Line 31: I find that "infer this mechanism to be at work in many extragalactic cases" is a little too strong. Better would be "suggest that this mechanism could also be at work" or "hypothesise that this mechanism could also be at work". This is not an inference, but a hypothesis/speculation.

Agreed. We opted for the the first “suggest that ...” sentence.

- Lines 73-78: As mentioned before, I would like to see a reference to Heinz (2002) in this discussion - either here, or possibly on line 236.

The reference to Heinz (2002) has been added on line 236 where it better fits and the following ones have been renumbered.

- Line 133: The jet alignment timescale estimates assume a constant mass accretion rate and disc alpha viscosity parameter. Please mention this assumption explicitly, as most black hole X-ray binary systems are transient (see Coriat et al. 2012, who note explicitly that the persistent

nature of GRS1758-258 is unclear). Should the source be a long duty cycle transient, the alignment timescale would be longer.

Ok. The assumption of constant mass accretion rate and alpha viscosity parameter is now stated explicitly.

- Line 248: "yields another remarkable implication" is again a little too strong, since it is followed by a reasonable remark about speculating further on the inferences. I would recommend toning that statement down.

Agreed. We lowered the tone by changing the word "remarkable" to "possible".

- Line 257: eMERLIN is a current instrument that is already available, and has been for several years.

Corrected. eMERLIN is now quoted among the existing interferometers.

Finally, we wish to express to Referee #2 our gratitude for his/her very constructive comments throughout the publication process that helped to improve our manuscript.